# LIGHTWEIGHT GRAPH-FREE CONDENSATION WITH MLP-DRIVEN OPTIMIZATION

## ABSTRACT

Graph condensation aims to compress large-scale graph data into a small-scale one, enabling efficient training of graph neural networks (GNNs) while preserving strong test performance and minimizing storage demands. Despite the promising performance of existing graph condensation methods, they still face two-fold challenges, i.e., *bi-level optimization inefficiency* & *rigid condensed node label design*, significantly limiting both efficiency and adaptability. To address such challenges, in this work, we propose a novel approach: **LIGHT**weight **G**raph-**F**ree **C**ondensation with MLP-driven optimization, named **LIGHTGFC**, which condenses large-scale graph data into a structure-free node set in a simple, accurate, yet highly efficient manner. Specifically, our proposed LIGHTGFC contains three essential stages: (S1) Proto-structural aggregation, which first embeds the structural information of the original graph into a proto-graph-free data through multi-hop neighbor aggregation; (S2) MLP-driven structural-free pretraining, which takes the proto-graph-free data as input to train an MLP model, aligning the structural condensed representations with node labels of the original graph; (S3) Lightweight class-to-node condensation, which condenses semantic and class information into representative nodes via a class-to-node projection algorithm with a lightweight projector, resulting in the final graph-free data. Extensive experiments show that the proposed LIGHTGFC achieves state-of-the-art accuracy across multiple benchmarks while requiring minimal training time (as little as 2.0s), highlighting both its effectiveness and efficiency.

## 1 INTRODUCTION

Graph Neural Networks (GNNs) have witnessed rapid development due to their strong learning capabilities for graph structural data (Kipf, 2016; Zheng et al., 2025; Liu et al., 2025; Xu et al., 2018a; Wu et al., 2019). Existing GNN models have been applied to extensive graph learning tasks and applications, such as social network analysis (Brody et al., 2021; Kipf, 2016; Wu et al., 2020), molecular representation (Xu et al., 2018a; Ying et al., 2021; 2018), transportation systems (Wang et al., 2024b;a), etc. However, in real-world scenarios, continual growth in the scale of graph data has led to a significant increase in storage and computational costs, posing a greater demand for efficient data condensation and processing techniques. To this end, graph condensation has been introduced to generate a small-scale synthetic graph from a large-scale original graph (Gao et al., 2025b; Jin et al., 2021; Li et al., 2023; Zheng et al., 2023; Liu et al., 2024; 2022; 2023), enabling GNNs trained on the condensed graph to serve as a compact substitute for the original, while preserving comparable performance on the original test graphs.

Existing mainstream graph condensation methods can be broadly categorized into three classes: gradient matching (Jin et al., 2021; Li et al., 2023), trajectory matching (Zheng et al., 2023; Liu et al., 2024), and distribution matching (Liu et al., 2022; 2023). Despite promising condensation performance, these matching-based methods either use gradients/trajectories of GNN training to match the learning behaviors of the

large-scale graph trained GNN with the condensed graph trained GNN, or match representation information to preserve the distribution consistency between the original graph and the condensed graph. However, while effective to some extent, such strategies still encounter two-fold challenges:

■ **C1: Complexity of bi-level optimization**, where condensed graph node features and structures, together with the newly trained GNN model, must be jointly optimized in a nested inner–outer loop. This requires repeatedly training and updating both the GNN and the condensed graph, resulting in low training efficiency and contradicting the objective of graph condensation.

■ **C2: Underestimation of condensed label design**, as existing methods typically adopt a predefined label distribution identical to the original graph (e.g., under a 10% condensation ratio with a three-class distribution of 60/20/20, 100 nodes are reduced to 10 nodes with three-class labels 6/2/2). However, such rigid preservation may not be optimal, since the label distribution of the condensed graph should ideally be refined according to the relative importance of nodes during condensation, rather than strictly mirroring the original node class distribution. In light of these, a natural question arises:

> **Question:** *Is it possible to deconstruct the complex bi-level optimization, while simultaneously considering the condensed graph data with class-aware node label importance?*

To answer this question and address these challenges, in this work, we propose a novel approach: **LIGHT**weight **G**raph-**F**ree **C**ondensation with MLP-driven optimization, named **LIGHTGFC**. Our method condenses large-scale graph data into a structure-free node set in a simple, accurate, and highly efficient manner. By deconstructing complex bi-level optimization with a single MLP-driven process, LIGHTGFC first performs data-centric structural condensation, and then learns a lightweight node feature projector that models condensed node label distributions through class-aware similarity. Specifically, our proposed LIGHTGFC contains three essential stages: (1) *Proto-structural aggregation*, which first embeds the structural information of the original graph into a proto-graph-free data through multi-hop neighbor aggregation, preventing loss of critical topology information in subsequent condensation; (2) *MLP-driven structural-free pretraining*, which takes the proto-graph-free data as input to train an MLP model, aligning the structural condensed representations with node labels of the original graph; (3) *Lightweight class-to-node condensation*, which condenses semantic and class information into representative nodes via a class-to-node projection algorithm. A lightweight projector is optimized using a prototype-aware feature alignment loss and a label-aware model adaptation loss, resulting in the final graph-free data. Extensive experiments on five widely used graph condensation benchmarks demonstrate that our method delivers strong condensation performance, achieving superior node classification accuracy while significantly reducing training time. In summary, our contributions are listed in threefold:

- **Lightweight Condensation**. We first propose a lightweight structure-free graph condensation framework, named **LIGHTGFC**, which condenses large-scale graphs into compact node sets via MLP-driven optimization, overcoming bi-level inefficiency and rigid label design while preserving structural semantics and class discriminability;

- **Three-stage Pipeline**. We design a concise three-stage pipeline for LIGHTGFC covering: S1: Proto-structural aggregation, to embed original topology into proto-graph-free data; S2: MLP-driven structure-free pretraining, to align condensed representations with original node labels, and S3: Lightweight class-to-node condensation, to generate representative nodes with adaptive optimization objectives;

- **Effectiveness & Efficiency.** Extensive experiments on five widely used graph condensation benchmarks demonstrate that LIGHTGFC achieves superior node classification (up to 8% improvement) accuracy while reducing training time (as little as 2.0s), with expressive performance and efficiency.

## 2 RELATED WORK

**Graph Condensation.** Mainstream graph condensation methods can be categorized into three groups: gradient matching (Jin et al., 2021; Yang et al., 2023), trajectory matching (Zheng et al., 2023; Zhang et al.,

2024), and distribution matching (Liu et al., 2022; Gao et al., 2025a). For gradient matching methods, typically, GCond (Jin et al., 2021) aligns the GNN training gradients on the original and synthetic graphs, and SGDD (Yang et al., 2023) uses the graphon approximation to implement optimal transport for Laplacian energy distribution matching. For trajectory matching methods, the pioneering method SFGC (Zheng et al., 2023), which creatively synthesizes graph-free data, matches the GNN training trajectories by scores. GEOM (Zhang et al., 2024) expands this idea by enforcing cross-graph consistency. For distribution matching methods, GCDM (Liu et al., 2022) takes the original feature matrix as a distribution over the receptive fields of nodes. CGC (Gao et al., 2025a) adopts a feature enhancement and condensation strategy to obtain graph-free data. GCPA (Li et al.) and DisCo (Xiao et al., 2025) both train the feature distribution of nodes, the former uses feature adaptation, and the latter relies on the MLP model after decoupling. ProStack (Bai et al., 2025) employs a graph memory mechanism to store feature distributions and topological structures for progressive condensation at different compression ratios. For other hybrid condensation methods, Bonsai (Gupta et al., 2024) introduces a computation tree to capture the diverse computational structures of GNNs; SNTK (Xu et al., 2023) leverages the Neural Tangent Kernel to guide the condensation process.

## 3 METHODOLOGY

**Notations.** The large-scale graph data, which needs to be condensed, is denoted as $\mathcal{G} = (\mathbf{X}, \mathbf{A}, \mathbf{Y})$, where $\mathbf{X} \in \mathbb{R}^{N \times d}$ denotes the $N$ number of nodes with $d$-dimensional features, $\mathbf{A} \in \mathbb{R}^{N \times N}$ denotes the adjacency matrix with the edge connections and $\mathbf{Y} \in \mathbb{R}^{N \times C}$ denotes the $C$-classes of node labels. In this paper, we propose a graph-free graph condensation paradigm designed to synthesize a compact set of graph nodes. The condensed graph is defined as $\mathcal{G}' = (\mathbf{X}', \mathbf{A}', \mathbf{Y}')$, where $\mathbf{X}' \in \mathbb{R}^{N' \times d}$ denotes the $N'$ number of nodes with $d$-dimensional features with $N' << N$ and $\mathbf{A}' \in \mathbb{R}^{N' \times N'}$ is the adjacency matrix of the condensed graph structure, and $\mathbf{Y}' \in \mathbb{R}^{N' \times C}$ denotes the $C$-classes of node labels. Existing methods leverage the predefined condensation ratio $r$ to synthesize the nodes of the condensation graph, $N' = \Sigma_{c=1}^{C}(r \cdot N_c)$, and $N_c = p_c \cdot N$ is the number of nodes for the $c$-th class in the original graph with the node class distribution proportion $p_c$.

**Graph Condensation with Bi-level Optimization.** Given a GNN model parameterized by $\boldsymbol{\theta}$, most existing graph condensation is defined as a bi-level optimization objective. Specifically, taking the original graph $\mathcal{G} = (\mathbf{X}, \mathbf{A}, \mathbf{Y})$ as input, the graph condensation requires simultaneously learning two objectives: (1) A condensation graph $\mathcal{G}' = (\mathbf{X}', \mathbf{A}', \mathbf{Y}')$ by aligning the learning behavior of the original graph trained $\text{GNN}_{\phi_{\mathcal{G}}}$ and the condensed graph trained $\text{GNN}_{\boldsymbol{\theta}^*_{\mathcal{G}'}}$ through a matching loss $\mathcal{L}_{\text{match}}(\cdot)$; (2) A new-trained $\text{GNN}_{\boldsymbol{\theta}^*_{\mathcal{G}'}}$ model for optimizing condensed graph class learning through a classification loss $\mathcal{L}_{\text{cls}}(\cdot)$. The formula of the bi-level problem is as follows:

$$\min_{\mathcal{G}'} \mathcal{L}_{\text{match}}\big[\text{GNN}_{\boldsymbol{\theta}^*_{\mathcal{G}'}}(\mathcal{G}), \text{GNN}_{\phi_{\mathcal{G}}}(\mathcal{G}')\big], \quad \text{s.t.} \quad \boldsymbol{\theta}^*_{\mathcal{G}'} = \arg\min_{\boldsymbol{\theta}_{\mathcal{G}'}} \mathcal{L}_{\text{cls}}\big[\text{GNN}_{\boldsymbol{\theta}_{\mathcal{G}'}}(\mathbf{X}', \mathbf{A}'), \mathbf{Y}'\big], \quad (1)$$

where $\boldsymbol{\theta}^*_{\mathcal{G}'}$ is the optimal parameters of the model trained on $\mathcal{G}'$. Furthermore, existing methods with graph structure condensation usually optimize a parameterized graph structure module $g(\cdot, \boldsymbol{\psi})$ through $\mathbf{A}' = \arg\min_{\boldsymbol{\psi}} \mathcal{L}_{\text{gsl}}\big[g(\mathbf{X}', \boldsymbol{\psi})\big]$. Consequently, the training process requires the simultaneous and iterative optimization of the GNN parameters $\boldsymbol{\theta}$, graph structure module parameter $\boldsymbol{\psi}$, and the condensed graph $\mathcal{G}'$.

### 3.1 OVERALL FRAMEWORK

Figure 2 presents the overall framework of our proposed LIGHTGFC, which comprises three essential stages: (S1) Proto-structural aggregation, (S2) MLP-driven structural-free pretraining, and (S3) Lightweight class-to-node condensation. Specifically, given the original large-scale graph data, we first sent it to S1, a proto-structural aggregation, to comprehensively aggregate multi-hop neighbors, thereby implicitly embedding the raw structural information into the proto-graph-free data. Then, the obtained proto-graph-free data would

Figure 1: Overall framework of the proposed light-weight graph-free condensation method LIGHTGFC.

be fed into S2, MLP-driven structural-free pretraining, where an MLP model is trained to compile with the structure-free condensation and align the structure-condensed representations with node labels of the original graph. After that, we conduct S3, a lightweight class-to-node condensation, which condenses the proto-graph-free data through similarity-based node feature mapping to synthesize class-aware graph-free data. A parameterized condensed feature projector is followed to map the class-aware graph-free node set under two constraints: label-aware model adaptation loss and a prototype-aware feature alignment loss, leading to the final condensed graph-free data.

## 3.2 PROTO-STRUCTURAL AGGREGATION

To address the complexity of bi-level optimization in graph condensation, we first propose to introduce the structure-free condensation through the proto-structural aggregation module, so that the original topological structure can be implicitly embedded within the graph-free node features. Given the original large-scale graph, $\mathcal{G} = (\mathbf{X}, \mathbf{A}, \mathbf{Y})$, we first capture structural information by aggregating multi-hop information from neighbor nodes through the typical graph convolutional operation as:

$$\hat{\mathbf{A}} = \tilde{\mathbf{D}}^{-\frac{1}{2}}(\mathbf{A} + \mathbf{I}_N)\tilde{\mathbf{D}}^{-\frac{1}{2}}, \quad \mathbf{H}_{k-1} = \hat{\mathbf{A}}^{(k)} \cdot \mathbf{X}, \tag{2}$$

where $\mathbf{I}_N$ is the identity matrix denoting self-connections and $\tilde{\mathbf{D}}$ is the degree matrix. By multi-hop topology aggregation, we get the aggregation feature matrix $\mathbf{H}_{k-1}$ ($k = 1, 2..., K$), where $k$ is the $k$-hop neighbor nodes. For the initial feature matrix, we have $\mathbf{H}_0 = \mathbf{X}$.

After the primitive capture of structural information in the original graph, we calculate the global average aggregation to obtain the *proto-graph-free data* $\mathcal{G}^0_{\text{sc}}$ with the balanced and comprehensive topological structure condensation, as follows:

$$\mathcal{G}^0_{\text{sc}} = (\mathbf{H}^0_{\text{sc}}, \mathbf{Y}), \text{ where } \mathbf{H}^0_{\text{sc}} = \frac{1}{K}\sum_{k}^{K} \mathbf{H}_{k-1}. \tag{3}$$

Here, $\mathbf{H}_{\text{sc}}^0 \in \mathbb{R}^{N \times d}$ and $K$ denotes the breadth of the neighbor nodes. In this stage, the proto-structural aggregation stage yields a structure-condensed node feature matrix that explicitly encodes topological information in the original graph.

### 3.3 MLP-DRIVEN STRUCTURAL-FREE PRETRAINING

After the proto-structural aggregation, we shift the attention from training GNN models to training an MLP model. When the proto-graph-free data $\mathcal{G}_{\text{sc}}^0$ has implicitly embedded the topological structure information, it still needs to be aligned with node labels in the original graph. In this process, it is necessary to preserve the classification ability of the condensed node features. In light of this, we propose to utilize the MLP as a structure-free model expert, enabling it to perform node classification without relying on graph structure. To achieve this goal, we propose an MLP-driven structural-free pretraining stage, where the MLP model parametrized by $\phi$ is trained to preserve the feature adaptability of $\mathcal{G}_{\text{sc}}^0$ through the supervision from node class labels $\mathbf{Y}$ in the original graph. The optimization objective can be defined as minimizing the cross-entropy loss $\mathcal{L}_{\text{ce}}$ as:

$$\min_{\phi} \mathcal{L}_{\text{ce}}(\text{MLP}_{\phi}(\mathbf{H}_{\text{sc}}^0), \mathbf{Y}). \tag{4}$$

### 3.4 LIGHTWEIGHT CLASS-TO-NODE CONDENSATION

Different from the existing node-centric condensation methods that directly allocate the number of nodes in the condensed graph according to the original label distribution, which can not fully reflect the importance relationship of the feature and structure for each class. We propose a stage of lightweight class-to-node condensation, which aims to adaptively adjust the class proportions based on their overall influence reflected in node labels. After this adjustment, the number of nodes assigned to each class is determined by its relative importance, ensuring that influential classes are better represented in the condensed graph. Hence, our class-to-node condensation balances the distribution of classes in quantitative proportion and structural importance, which leads to a more faithful preservation of class semantics in the condensed graph-free data.

Specifically, we calculate the feature discrepancy between our derived $\mathbf{H}_{\text{sc}}^0 \in \mathcal{G}_{\text{sc}}^0$ and the original graph $\mathbf{X} \in \mathcal{G}$ as follows:

$$w^c = \sum_{\mathbf{x}_i \in \mathcal{V}_c} w_i^c, \text{ where } w_i^c = 1 - \text{Dist}\left[\mathbf{x}_i^c, \mathbf{h}_{(i,\text{sc})}^{0,c}\right], \tag{5}$$

where $\mathbf{x}_i \in \mathbf{X}$ and $\mathbf{h}_{(i,\text{sc})}^0 \in \mathbf{H}_{\text{sc}}^0$ denoting the original graph node feature and the proto-graph-free data node feature, respectively. $\mathcal{V}_c$ is defined as the set of nodes in the graph which belong to class $c$. Moreover, $w_c \in \mathbb{R}^1$ is a scalar, denoting the class-aware similarity as the weight value for each class $c$, and $\text{Dist}[\cdot, \cdot]$ is the discrepancy metric function. In our work, we use the cosine similarity $\cos(\cdot)$. Such similarity measures the degree of variation in node features introduced by our proposed proto-structural condensation, relative to the original graph topology. Nodes with high $w_i^c$ capture richer structural information from their neighbors, implying that they are more central and informative within the original graph.

Such class-aware similarity determines the proportion of nodes of that class in the condensed graph with $|\mathcal{V}_c| = N_c' = w_c \cdot N'$. For each class, the condensed graph-free feature vector is aggregated by the features of nodes in the proto-graph-free data according to their respective weights, so we have:

$$\mathbf{h}_{(j,\text{cg})}^c = \sum_{i=1}^{(p_c \cdot N)/(w_c \cdot N')} \frac{w_i^c}{w_c} \cdot \mathbf{h}_{(i,\text{sc})}^0, \quad \mathbf{H}_{\text{cg}} = \left\{\mathbf{h}_{(j,\text{cg})}^c \,\Big|\, j = 1, \ldots, N_c', \ c = 1, \ldots, C\right\}, \tag{6}$$

Where $(p_c \cdot N)/(w_c \cdot N')$ denotes the number of condensed graph nodes obtained by aggregating the nodes of the original graph for $c$-th class. Therefore, we obtain the initial feature matrix of condensed graph-free

data $\mathbf{H}_{\text{cg}} \in \mathbb{R}^{N' \times C}$. Meanwhile, we obtain the node labels $y'_j = c, j \in N'_c$, according to the weighted node class distribution. Furthermore, the label vector of the condensed graph-free data can be synthesized $\mathbf{Y}' = \left\{ y'_j \right\}_{j=1}^{N'_c} \in \mathbb{R}^{N' \times 1}$. In this way, we obtain the initial condensed graph-free data $\mathcal{G}_{\text{sc}}^1 = (\mathbf{H}_{\text{cg}}, \mathbf{Y}')$.

Although $\mathbf{H}_{\text{cg}}$ preserves structural aggregation from the original graph, it is still limited by the initial condensed representation from two aspects: (a) it may not align well with the semantic space required for downstream learning, and (b) it lacks discriminative power since class information is not explicitly optimized. To address this, we introduce a *lightweight projector*, parametrized by a learnable matrix $\mathbf{M}$, to further map $\mathbf{H}_{\text{cg}}$ into a task-adaptive feature space, so that we obtain the final condensed graph-free data:

$$\mathcal{G}' = (\mathbf{X}', \mathbf{Y}'), \text{ where } \mathbf{X}' = \mathbf{P} \cdot \mathbf{H}_{\text{cg}}. \tag{7}$$

This projector is optimized with a prototype-aware feature alignment and label-aware model adaptation losses, to further enhance the semantic consistency and improve the effectiveness of the final condensed graph-free data.

■ *Label-aware Model Adaption Loss.* To ensure that the explicit information embedded from the original graph by the pretrained model $\text{MLP}_\phi^*$ is effectively transferred to the condensed graph, we design a model adaptation loss as:

$$\mathcal{L}_{\text{adapt}} = \text{Cross-entropy}(\text{MLP}_\phi^*(\mathbf{X}'), \mathbf{Y}'). \tag{8}$$

■ *Prototype-aware Feature Alignment Loss.* To mitigate information loss during this process and maintain the original feature distribution, the feature vector of each condensed node $\mathbf{x}'_i \in \mathcal{G}'$ is enforced to align with the class mean feature from the original graph $\mathbf{x}_i \in \mathcal{G}$. Based on this principle, we have:

$$\mathcal{L}_{\text{align}} = \sum_{c=1}^{C} \left\| \frac{1}{N'_c} \sum_{i:y'_i=c} \mathbf{x}'_i - \frac{1}{N_c} \sum_{i:y_i=c} \mathbf{x}_i \right\|_2^2. \tag{9}$$

Therefore, the total loss function is,

$$\mathcal{L}_{\text{total}} = \alpha \cdot \mathcal{L}_{\text{adapt}} + \beta \cdot \mathcal{L}_{\text{align}}, \tag{10}$$

where $\alpha$ and $\beta$ are hyper-parameters to control the weights of two optimization objectives.

## 4 EXPERIMENTS

We evaluate the proposed LIGHTGFC for the condensation performance on the node classification, generalization ability, complexity and efficiency, as well as the ablation study in terms of each submodule in LIGHTGFC. Specifically, our objective is to answer the following questions: **Q1**: What is the condensation performance of our proposed LIGHTGFC compared with existing baseline methods? **Q2**: How well does our LIGHTGFC perform on different GNN architectures in terms of generalization ability? **Q3**: In terms of condensation time consumption and memory usage, does LIGHTGFC achieve good performance with lightweight efficiency? **Q4**: What is the performance of each submodule within our LIGHTGFC in the ablation study? **Q5**: How sensitive are the hyperparameters that influence the performance of LIGHTGFC?

### 4.1 EXPERIMENTAL SETUP

**Datasets & Baselines.** We use widely used node-level graph condensation datasets covering: Three transductive datasets, Cora, Citeseer (Kipf, 2016) and Ogbn-Arxiv (Hu et al., 2020), and two inductive datasets, Flickr (Zeng et al., 2019) and Reddit (Lee et al., 2009). Detailed statistics are summarized in Appendix A.2.

Table 1: Node classification accuracy (ACC±std%) comparison between our LIGHTGFC *vs.* baseline methods on different datasets under various condensation ratios. The best results are bold, and the second-best results are underlined. 'OOM' indicates out-of-memory on NVIDIA 4090D GPU with 24 GB.

| Datasets | Ratios | GCond-X | GCDM-X | SNTK-X | SFGC | GEOM | CGC-X | GCPA | DisCo | ProStack | Bonsai | LIGHTGFC (ours) | Whole Datasets |
|---|---|---|---|---|---|---|---|---|---|---|---|---|---|
| Cora | 1.30% | 75.9±1.2 | 81.3±0.4 | 82.2±0.5 | 80.1±0.4 | 80.3±1.1 | _83.4±0.3_ | 82.1±0.6 | 76.9±0.8 | 82.5±0.0 | 83.2±0.3 | **89.0±0.3** | 81.2±0.2 |
| | 2.60% | 75.7±0.9 | 81.4±0.1 | 82.4±0.5 | 81.7±0.5 | 81.5±0.8 | 83.4±0.4 | 82.9±1.0 | 78.7±0.3 | 83.0±0.0 | _84.6±0.2_ | **90.6±0.9** | |
| | 5.20% | 76.0±0.9 | 82.5±0.3 | 82.1±0.1 | 81.6±0.8 | 82.2±0.4 | 82.8±0.0 | 82.3±0.7 | 78.8±0.5 | 83.2±0.0 | _85.5±0.7_ | **89.8±0.4** | |
| Citeseer | 0.90% | 71.4±0.8 | 69.0±0.5 | 69.9±0.4 | 71.4±0.5 | 71.1±0.2 | 72.1±0.2 | 75.4±0.4 | 70.2±0.3 | 72.4±0.0 | _76.5±0.7_ | **81.8±0.3** | 71.7±0.1 |
| | 1.80% | 69.8±1.1 | 71.9±0.5 | 69.9±0.5 | 72.4±0.4 | 71.3±0.1 | 72.6±0.2 | 74.8±0.3 | 71.6±0.5 | 72.7±0.0 | _77.1±0.2_ | **82.8±0.3** | |
| | 3.60% | 69.4±1.4 | 72.8±0.6 | 69.1±0.4 | 70.6±0.7 | 72.1±1.0 | 71.4±0.4 | 74.9±0.1 | 72.1±0.1 | 73.1±0.0 | _75.6±0.5_ | **83.2±0.5** | |
| Arxiv | 0.05% | 61.3±0.5 | 61.0±0.1 | 63.9±0.3 | 65.5±0.7 | 64.7±0.4 | 64.0±0.1 | _67.2±0.3_ | 64.0±0.7 | 65.2±0.0 | 59.6±0.6 | **67.6±0.2** | 71.4±0.1 |
| | 0.25% | 64.2±0.4 | 61.2±0.1 | 65.5±0.1 | 66.1±0.4 | 67.5±0.3 | 66.3±0.3 | 67.7±0.2 | 65.9±0.5 | _68.0±0.0_ | 58.9±0.7 | **68.1±0.3** | |
| | 0.50% | 63.1±0.5 | 62.5±0.1 | 65.7±0.4 | 66.8±0.4 | 67.6±0.2 | 67.0±0.1 | _68.1±0.1_ | 66.2±0.1 | **68.9±0.0** | 66.1±0.1 | 67.7±0.1 | |
| Flickr | 0.10% | 45.9±0.1 | 46.0±0.1 | 46.6±0.3 | 46.6±0.2 | 46.1±0.5 | 46.7±0.2 | _47.2±0.3_ | 46.2±0.4 | - | 46.2±0.5 | **47.2±0.5** | 47.2±0.1 |
| | 0.50% | 45.0±0.2 | 45.6±0.1 | 46.7±0.1 | 47.0±0.1 | 46.2±0.2 | 47.0±0.1 | 47.1±0.1 | 47.0±0.1 | - | _47.3±0.4_ | **47.3±0.5** | |
| | 1.00% | 45.0±0.1 | 45.4±0.3 | 46.6±0.2 | 47.0±0.1 | 46.7±0.1 | 47.0±0.1 | _47.2±0.1_ | 46.8±0.3 | - | 46.9±0.0 | **47.4±0.2** | |
| Reddit | 0.05% | 88.4±0.4 | 86.5±0.2 | OOM | 89.7±0.2 | 90.1±0.2 | 90.3±0.2 | 90.5±0.3 | 91.4±0.2 | _92.0±0.0_ | 82.3±0.3 | **92.1±0.3** | 93.9±0.0 |
| | 0.10% | 89.3±0.1 | 87.2±0.1 | OOM | 90.0±0.2 | 90.4±0.1 | 90.8±0.2 | **93.0±0.1** | 91.8±0.3 | _92.4±0.0_ | 86.1±0.1 | 91.8±0.4 | |
| | 0.20% | 88.8±0.4 | 88.8±0.1 | OOM | 89.9±0.4 | 90.9±0.1 | 91.4±0.1 | **92.9±0.2** | 91.7±0.3 | _92.7±0.0_ | 88.2±0.6 | 91.7±0.7 | |

Table 2: The generalization ability comparison between baseline methods and LIGHTGFC.

| Datasets | Models | GCOND | SFGC | GCDM | DisCo | SGDD | CGC | LIGHTGFC (ours) |
|---|---|---|---|---|---|---|---|---|
| Cora ($r = 2.6\%$) | MLP | 73.1 | 81.1 | 69.7 | 59.5 | 76.8 | 70.7 | **81.6** |
| | GCN | 80.1 | 81.1 | 77.2 | 78.6 | 79.8 | 83.2 | **89.3** |
| | SAGE | 78.2 | 81.9 | 73.4 | 75.6 | 80.4 | 67.7 | **82.7** |
| | SGC | 79.3 | 79.1 | 75.0 | 75.0 | 78.5 | 79.9 | **89.1** |
| | GIN | 66.5 | 72.9 | 63.9 | 74.2 | 72.8 | 46.7 | **80.6** |
| | JKNet | 80.7 | 79.9 | 77.8 | 78.7 | 76.9 | 81.3 | **88.9** |
| Ogbn-arxiv ($r = 0.5\%$) | MLP | 43.8 | 46.6 | 41.8 | 49.5 | 36.9 | 40.9 | **50.3** |
| | GCN | 64.0 | 66.8 | 61.7 | 66.2 | 65.6 | 64.0 | **67.5** |
| | SAGE | 55.9 | **63.8** | 53.0 | 64.2 | 53.9 | 47.9 | 59.2 |
| | SGC | 63.6 | 63.8 | 60.1 | 64.9 | 62.2 | 63.9 | **66.4** |
| | GIN | 60.1 | 61.9 | 58.4 | 63.2 | 59.1 | 59.3 | **64.9** |
| | JKNet | 61.6 | 65.7 | 57.2 | 66.2 | 60.1 | 62.6 | **66.9** |
| Reddit ($r = 0.2\%$) | MLP | 48.4 | 45.4 | 40.5 | 44.8 | **55.2** | 42.9 | 49.3 |
| | GCN | 91.7 | 87.8 | 83.3 | 92.6 | 91.8 | 89.6 | **92.7** |
| | SAGE | 73.0 | 77.9 | 55.0 | 84.4 | **89.0** | 71.9 | 85.9 |
| | SGC | 92.2 | 87.6 | 79.9 | 92.3 | 92.5 | 91.0 | **92.3** |
| | GIN | 83.6 | 80.3 | 78.8 | 88.9 | 85.4 | 82.0 | **90.1** |
| | JKNet | 87.3 | 88.2 | 77.3 | 91.3 | 90.5 | 85.5 | **91.0** |

We compare our approach with the baseline graph condensation methods for node classification, which mainly cover two categories: (1) Graph condensation methods, including GEOM (Zhang et al., 2024), Bonsai (Gupta et al., 2024), ProStack (Bai et al., 2025), and DisCo (Xiao et al., 2025); and (2) Condensation methods for graph-free data, including: GCond-X (Jin et al., 2021), GCDM-X (Liu et al., 2022), SNTK-X (Xu et al., 2023), SFGC (Zheng et al., 2023), CGC-X (Gao et al., 2025a), GCPA (Li et al.). Following GCond (Jin et al., 2021), we set three compression ratios for each dataset to conduct experiments. We evaluate the generalization ability in various model architectures, including an MLP and four more distinct GNN models: SAGE (Hamilton et al., 2017), SGC (Wu et al., 2019), GIN (Xu et al., 2018a), and JKNet (Xu et al., 2018b). The average node classification accuracy (ACC%) and the corresponding standard deviation (±std%) in 10 runs are reported.

## 4.2 OVERALL CONDENSATION PERFORMANCE

In Table 1, the node classification performance of the graphs condensed by LIGHTGFC and the baseline methods. In general, we could observe that our LIGHTGFC achieves the highest prediction accuracy in 12

Table 3: Ablation components ($\checkmark \times$) and performance. 'PAlign' denotes the prototype-aware feature alignment, 'LAdapt' denotes the label-aware model adaption, and 'SC' denotes the node-class similarity-based condensation. Best results are in bold.

| Variants | Idx0 (**Ours.**) PAlign / LAdapt / SC $\checkmark \checkmark \checkmark$ | Idx1 PAlign / LAdapt / SC $\checkmark \times \times$ | Idx2 PAlign / LAdapt / SC $\times \checkmark \checkmark$ | Idx3 PAlign / LAdapt / SC $\times \times \checkmark$ | Idx4 PAlign / LAdapt / SC $\checkmark \checkmark \times$ |
|---|---|---|---|---|---|
| Cora ($r = 2.60\%$) | **89.0** | 86.3 | 76.1 | 66.2 | 80.2 |
| Citeseer ($r = 1.80\%$) | **82.9** | 78.3 | 63.3 | 53.0 | 68.9 |
| Arxiv ($r = 0.25\%$) | **67.4** | 62.1 | 60.5 | 58.4 | 60.7 |
| Flickr ($r = 0.5\%$) | **47.0** | 45.9 | 44.5 | 43.3 | 45.0 |
| Reddit ($r = 0.10\%$) | **92.4** | 72.8 | 59.1 | 56.9 | 66.7 |

cases out of the 15 condensation scenarios and datasets, which demonstrates the superior effectiveness of our proposed LIGHTGFC. Specifically, compared with graph-free condensation methods (SFGC, CGC-X, and GCPA), our proposed LIGHTGFC delivers stronger results by aggregating and preserving richer topological information within condensed nodes, where the advantage stems from its proto-structural aggregation submodule. Relative to other latest methods (DisCo, ProStack, and Bonsai), our method exhibits stronger stability across multiple datasets. LIGHTGFC achieves SOTA in four datasets (Cora, Citeseer, Arxiv, and Flickr), while both Bonsai and ProStack only achieve the second-best performance on two datasets.

Especially in the Cora and Citeseer dataset of all ratio, LIGHTGFC achieves the SOTA performance and outperforms the second-highest accuracy by 5% (Bonsai 84.6% vs. ours 90.6%). The two datasets have a more complex feature matrix, which not only can maintain efficient training of feature distributions, but also can accommodate more feature alignment and model adaption information, leading to the significant improvement of our proposed method. It shows the powerful representation capabilities of the data condensed by LIGHTGFC, especially in complex feature graph datasets. In the Flickr dataset, the situation is exactly the opposite. Simple feature matrix have limited capacity to accommodate sophisticated information, so the improvement is quite limited. For Ogbn-Arxiv and Reddit, which have more classes, the class-to-node condensation plays a key role in balancing the synthesis process of each class. In summary, the experimental results above show that our proposed method could balance the influence of feature and topology information in graph-free data on the downstream classification task.

### 4.3 GENERALIZATION ABILITY AND EFFICIENCY

**Generalization Ability.** As shown in Table 2, we train a diverse set of model architectures on condensed data condensed by LIGHTGFC and the baseline methods, including GCN, SAGE, SGC, GIN and JKNet, and evaluate the original test graph node classification performance. The experimental results demonstrate that the graphs produced by LIGHTGFC exhibit a better generalization performance compared to the baseline methods in most GNN architectures. These GNN architectures have different focus. GCN and SGC require the graph to possess discriminative features; GIN and JKNet are necessary to preserve local or more global topological information. So, it proves that our proposed LIGHTGFC achieves a good balance in features, local and global structures, and multi-scale information. In summary, we attribute the strong generalization capability of our proposed LIGHTGFC to our class-to-node condensation process. Compared to the previous baseline approach, the class-to-node condensation we proposed can aggregate feature information by the similarity between a certain node and its class label, which improves the generalization robustness.

**Efficiency.** We compare the condensation time consumption (in seconds) of LIGHTGFC and the baseline methods with NVIDIA 4090D GPU in 500 epochs. From the observation of Figure 2a, LIGHTGFC significantly outperforms conventional methods, achieving the best time efficiency in all five datasets. Compared to others, the increase in the condensation time of LIGHTGFC remains moderate as the data scale expands (from Arxiv to Flickr and Reddit). We also compare the dynamic memory analysis of our method and other approaches: GEOM, which represents distribution matching; GCPA, which reflects the graph-free conden-

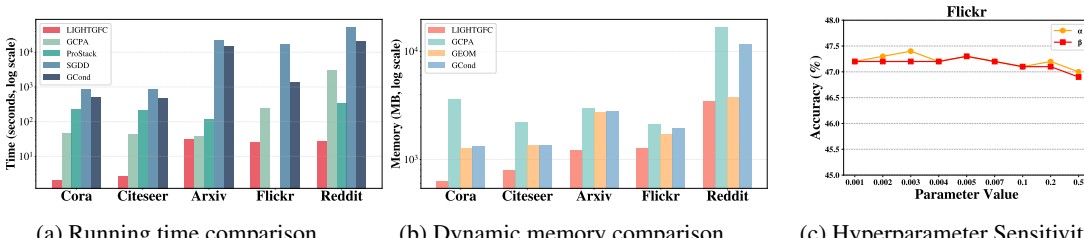

(a) Running time comparison.  (b) Dynamic memory comparison.  (c) Hyperparameter Sensitivity.

Figure 2: Overall comparison of condensation time, memory, and sensitivity across datasets.

sation method; and GCond, the most representative bi-level compression method. In Figure 2b, LIGHTGFC achieves the best performance in all datasets, which significantly saves more than half memory in Cora, Citeseer, Arxiv, and Reddit. Our graph-free method only trains on the node feature matrix, using a single variable reduces memory consumption. We attribute the efficiency of our proposed LIGHTGFC to proto-structural aggregation submodule, which embeds topological information within the feature matrix, eliminates the extensive running time costs, leading to significant acceleration in speed.

### 4.4 IN-DEPTH EXPERIMENTAL ANALYSIS

**Ablation Study.** To validate the effectiveness of each component within LIGHTGFC, we disabled individual loss components and replaced the condensation method in the ablation study. As detailed in Table 3, we created several variants of the model (Idx1-Idx3), which individually disable the components of the prototype-aware feature alignment and the label-aware model adaption. For Idx4, we replaced the similarity class-to-node condensation with the standard K-Center method. Given these results, we have the following essential observations. First, prototype-aware feature alignment has a significant impact, which preserves the original graph topological information and feature discriminability within the condensed graph-free data. Second, label-aware model adaption has been proven to be a vital component for maintaining a balanced feature distribution for each class in the class-to-node condensation. Finally, the results confirm that the similarity class-to-node condensation serves as the fundamental basis of LIGHTGFC, upon which both loss optimization modules are built to achieve optimal performance.

**Hyperparameter Analysis.** To determine the influence of hyperparameter on experimental precision, we analyze two essential hyperparameters, $\alpha$ and $\beta$ adjust the scale of total optimization loss in Eq.( 10). As illustrated in the corresponding Figure 2c, we selected the most representative dataset and performed multiple sets of experiments with varying hyperparameter configurations. The results indicate that, while extreme values can affect the outcome, the model performance remains relatively stable in a wide range of settings. It demonstrates that the modules of feature adaptation and feature alignment have stable feature matrix optimization, which shows a stronger hyper-parameter robustness to sensitivity of LIGHTGFC.

## 5 CONCLUSION

In this work, we introduced LIGHTGFC, a lightweight structure-free graph condensation framework that effectively addresses the inefficiency of bi-level optimization and the rigidity of condensed label design. By decomposing the condensation process into three stages—proto-structural aggregation, MLP-driven structure-free pretraining, and lightweight class-to-node condensation—LIGHTGFC provides a simple yet powerful pipeline for generating compact graph-free data. Extensive experiments on multiple benchmarks demonstrate that LIGHTGFC achieves state-of-the-art accuracy while drastically reducing training time, highlighting both its effectiveness and efficiency. In future work, we aim to extend the applicability of graph condensation tailored to different downstream graph learning tasks, e.g., graph classification and link prediction, or diverse graph data types, e.g., multi-relational graphs and dynamic graphs.

## ETHICS STATEMENT

This work complies with the ICLR Code of Ethics. And we promises that our study does not involve human subjects, personally identifiable information, or sensitive data. All datasets used are publicly available and commonly employed in prior research, which can download on PyTorch Geometric (PyG) (Fey & Lenssen, 2019) and DGL (Wang et al., 2019). We have carefully considered potential ethical concerns, including fairness, bias, privacy, and possible misuse of our proposed method. We encourage responsible use of our approach within appropriate research contexts.

## REPRODUCIBILITY STATEMENT

We are committed to ensuring the reproducibility of our results. Specifically, we provide following: (1) Experimental settings: GNN model architectures, and training procedures are documented in Section 4.1; (2) Software and hardware: we use Intel(R) Core(TM) i9-14900KF CPU and NVIDIA 4090D GPU; And Linux (Ubuntu 20.04.6 LTS (GNU/Linux 5.15.0-139-generic x86 64)) with PyTorch Version 1.13.1+cu117 and PyTorch Geometric Version 2.6.1. To ensure the reproducibility of this work, we will release the code after this work is accepted.

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

# A APPENDIX

This is the Appendix of submission 'Lightweight Graph-Free Condensation With MLP-Driven Optimization'. In this appendix, we provide dataset statistics, more visualization results, as well as time complexity analysis.

## A.1 USE OF LLMS

During the writing process of this paper, we conservatively employed the Large Language Models (LLMs) exclusively for improving grammar, readability, and formatting. We guarantee that LLMs had no involvement in any technical content, including problem formulation, theoretical results, algorithms, and experiments, were entirely designed, implemented, and verified by the authors. No parts of the research ideas, results, or analysis were generated by an LLM.

## A.2 DATASET STATISTICS

We provide details of the statistics of the original dataset in Table 4.

Table 4: Details of dataset statistics.

| Dataset | Train/Val/Test Nodes | Nodes | Edges | Features | Classes |
| --- | --- | --- | --- | --- | --- |
| Cora | 140/500/1,000 | 2,708 | 5,429 | 1,433 | 7 |
| CiteSeer | 120/500/1,000 | 3,327 | 4,732 | 3,703 | 6 |
| Ogbn-arxiv | 90,941/29,799/48,603 | 169,343 | 1,166,243 | 128 | 40 |
| Flickr | 44,625/22,312/22,313 | 89,250 | 899,756 | 500 | 7 |
| Reddit | 153,431/23,831/55,703 | 232,965 | 57,307,946 | 602 | 41 |

## A.3 VISUALIZATION ANALYSIS

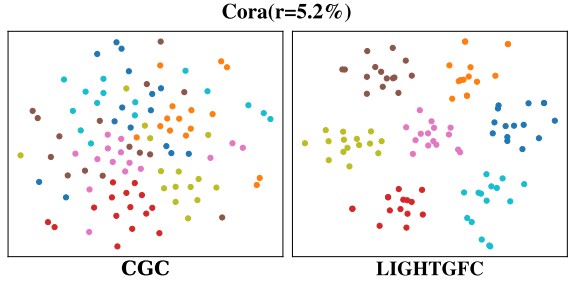

Figure 3: The T-SNE visualization of CGC and LIGHTGFC in Cora.

We present a T-SNE visualization of the data condensed by both the CGC model and our LIGHTGFC model on the Cora dataset (ratio = 5.2%) in Figure 3. The visual comparison of the two shows that the data produced by LIGHTGFC exhibit a much clearer and more distinct clustering distribution. It shows that our method can effectively capture high-quality node features and preserve implicit topological structures, which directly translates to its superior performance on downstream tasks.

## A.4 TIME COMPLEXITY ANALYSIS

The pipeline of LIGHTGFC consists of two core stages: proto-structural aggregation and lightweight class-to-node condensation. The time complexity analysis is derived accordingly.

According to the graph theory, the original graph can be denoted as $\mathcal{G} = (\mathcal{V}, \mathcal{E})$, where $\mathcal{V}$ is the set of graph nodes and $\mathcal{E}$ is the set of graph edges, the number of nodes is $N_v = |\mathcal{V}|$, the number of edges is $N_e = |\mathcal{E}|$, with $d$-dimensional feature. The number of nodes in the condensed graph is $N_v'$, where $N_v' << N_v$.

In proto-structural aggregation, LIGHTGFC aims to embed topological information into node features. Each node representation is updated by aggregating features from its neighbors, which incurs a time complexity of $O(N_e \cdot d)$. LIGHTGFC performs this operation only once as an initialization step, avoiding repetitive processing of the graph structure during condensation. Hence, the time complexity of this stage is $O(N_e \cdot d)$.

In lightweight class-to-node condensation, after obtaining structure-aware node features, LIGHTGFC reformulates the task as condensing a graph-free dataset. The core operations involve computing node weights. Specifically, the similarity between the original and aggregated features is calculated for all $N_v$ nodes, yielding a complexity of $O(N_v \cdot d)$. Node weights are derived, and weighted feature aggregation is performed within each class, which is again dominated by $O(N_v \cdot d)$.

By combining the two stages, the total time complexity of LIGHTGFC is: $O((N_v + N_e) \cdot d)$.

In most graphs the number of edges $N_e$ is typically larger than the number of nodes $N_v$, the complexity can be simplified to: $O(N_e \cdot d)$.

The efficiency of LIGHTGFC is evident in this linear complexity, which scales proportionally with the graph size (in terms of both nodes and edges). Moreover, by decoupling topology encoding from the condensation step, LIGHTGFC achieves substantial efficiency gains. The one-time feature aggregation replaces repeated adjacency matrix operations during iterative optimization, allowing the condensation process to be conducted entirely in feature space, thereby enabling faster execution and greater scalability.

