# OpenReview forum: "Lightweight Graph-Free Condensation with MLP-Driven Optimization"
_ICLR.cc/2026/Conference — Submitted to ICLR 2026_

### Official Review · Reviewer_j43U · 2025-10-27

**Soundness:** 2
**Presentation:** 2
**Contribution:** 2
**Rating:** 4
**Confidence:** 4

**Summary:**

This paper proposes a novel approach, LIGHTGFC, to address challenges in bilevel optimization and rigid node label distribution for graph condensation. Experiments on five standard benchmarks demonstrate that LIGHTGFC achieves superior performance compared to existing methods.

**Strengths:**

1. The proposed strategy to break the rigid label distribution constraint is a novel contribution to the field of graph condensation.
2. The paper provides a comprehensive empirical evaluation, comparing both performance and efficiency against a wide range of baselines.

**Weaknesses:**

1. The analysis in Figure 2(c) is confusing. The trend indicates that performance improves as $\alpha$ and $\beta$ decrease, suggesting the components S2 and S3 (which $\alpha$ and $\beta$ weight) may be unnecessary or even detrimental.
    - Can the authors explain this observation?
    - Why was only the Flickr dataset used for this specific analysis?
2. The paper claims, "Nodes with high $w_c^i$ capture richer structural information... implying that they are more central." This assumption is not sufficiently justified. While high *degree* is intuitively linked to centrality, the relationship between $w_c^i$ and centrality is not obvious. Please provide a theoretical explanation or experimental results (e.g., a correlation study between $w_c^i$ and node degree) to support this claim.
3. For a method that emphasizes efficiency, the evaluation lacks large-scale datasets. To robustly validate the scalability claims, it is essential to include benchmarks like Ogbn-products.
4. Given the significant performance gains reported, the paper must provide the source code or, at minimum, detailed experimental settings and the best hyperparameter configurations to ensure reproducibility.
5. Using 'Dist' (distance) to measure similarity is counter-intuitive, as distance and similarity are inversely related. Using $1 - \text{Dist}$ or an alternative formulation would be clearer.
6. The notation is unclear in places. The authors must explicitly define the subscripts $sc$ in $H_{sc}$ and $cg$ in $H_{cg}$.

## Minor:
1. **Terminology:** The terms "structural-free" and "graph-free" are used interchangeably. Please unify this terminology throughout the manuscript.
2. **Citation Errors:**
    - Line 102: The citation for GCPA appears to be incorrect.
    - Line 106: "SNTK" should be corrected to its official name, "KIDD."
3. **Clarity (Line 323):** Please clarify whether "10 rounds" refers to 10 *condensation* rounds or 10 *downstream GNN training* rounds.
4. **Typo (Line 255):** A space is missing after G'.
5. **Missing References:** The related work section should include and discuss the provided references [1]-[5], which cover relevant surveys, benchmarks, and other lightweight condensation methods.

## References:

[1] A Comprehensive Survey on Graph Reduction: Sparsification, Coarsening, and Condensation, In IJCAI 2024

[2] GC-Bench: An Open and Unified Benchmark for Graph Condensation, In NeurIPS 2024

[3] GC4NC: A Benchmark Framework for Graph Condensation on Node Classification with New Insights, In NeurIPS 2025

[4] Scalable graph condensation with evolving capabilities, arxiv 2025

[5] Simple Graph Condensation, In PKDD 2024

**Questions:**

See Weaknesses above.

---

> ### Author Response · Authors · 2025-11-21
> **Response(1) to Reviewer j43U**
>
> Thank you for the constructive review. We appreciate your positive comments regarding our method’s motivation. We have responded to each of your points in detail below.
>
> **[Re-Weakness (1)] The trend indicates that performance improves as $\alpha$ and $\beta$ decrease & without others dataset hyperparameter sensitivity experiment**
>
> > (1) Due to the strict page limits of ICLR, we were only able to include the results for the Flickr dataset, and the line graph of experimental results for the remaining datasets will be provided in the appendix of the revised version.
> >
> > Here are tables of hyperparameter sensitivity experiment:
>
> | Dataset  | Hyper. | 0.1  | 0.2  | 0.3  | 0.4  | 0.5  | 0.6  | 0.7  | 0.8  | 0.9  |
> | -------- | :----: | ---- | ---- | ---- | ---- | ---- | ---- | ---- | ---- | ---- |
> | Cora     |   α    | 88.0 | 87.0 | 86.6 | 87.3 | 87.7 | 87.2 | 87.0 | 87.0 | 87.3 |
> | (2.60%)  |   β    | 86.9 | 87.3 | 86.7 | 87.9 | 87.7 | 87.4 | 88.0 | 86.6 | 86.0 |
> | Citeseer |   α    | 82.5 | 82.6 | 82.7 | 82.5 | 82.7 | 82.6 | 82.6 | 82.6 | 82.6 |
> | (1.80%)  |   β    | 82.6 | 82.6 | 82.6 | 82.6 | 82.7 | 82.7 | 82.7 | 82.6 | 82.7 |
> | Arxiv    |   α    | 68.8 | 68.2 | 67.7 | 67.9 | 67.9 | 67.9 | 67.9 | 68.0 | 67.9 |
> | (0.25%)  |   β    | 68.4 | 67.9 | 68.0 | 67.9 | 67.9 | 68.0 | 67.9 | 68.3 | 67.6 |
>
> | Dataset | Hyper. | 0.001 | 0.002 | 0.003 | 0.004 | 0.005 | 0.007 | 0.1  | 0.2  | 0.5  |
> | ------- | :----: | ----- | ----- | ----- | ----- | ----- | ----- | ---- | ---- | ---- |
> | Flickr  |   α    | 47.2  | 47.3  | 47.4  | 47.3  | 47.2  | 47.1  | 47.2 | 47.2 | 47.0 |
> | (1.00%) |   β    | 47.2  | 47.2  | 47.2  | 47.3  | 47.2  | 47.1  | 47.2 | 47.1 | 46.9 |
> | Reddit  |   α    | 91.0  | 91.3  | 92.0  | 91.9  | 91.8  | 91.6  | 91.1 | 91.0 | 91.0 |
> | (0.10%) |   β    | 91.0  | 91.0  | 91.0  | 91.1  | 91.1  | 91.1  | 91.0 | 91.2 | 91.2 |
>
> > (2) From the observations of the hyperparameter sensitivity experiment results, we can conclude that: (i) the performance drop on the Flickr dataset is an exception, other datasets performed well; (ii) apart from some fluctuations in the alpha parameter on Cora and Reddit, the variations are generally small, demonstrating the robustness and strong performance of the method.
>
> > (3) For Flickr, the performance occurs within the parameter range of 0.007 to 0.5, where performance is inversely proportional to parameters(still includes segments of improvement).The more pronounced decline becomes visible after the value of 0.1. This volatility to the larger interval step size used in this latter range compared to the range of 0.001 to 0.005 , which naturally makes the results more prone to fluctuation.
>
> ------
>
> **[Re-Weakness (2)] Certification the influence of structural information**
>
> > To validate the statement “Nodes with high degree capture richer structural information, implying that they are more central,” we conducted an experiment on the Cora dataset. Specifically, we created three variants:
> >
> > - removing the top 10% highest-degree nodes,
> >
> > - removing the bottom 10% lowest-degree nodes,
> >
> > - and randomly removing 10% of the nodes.
> >
> > We then **trained a GCN on these three variants** as well as the original dataset and compared the results.
>
> | Dataset  | Whole Dataset | Remove Top10% | Remove Bottom10% | Random Remove10% |
> | -------- | :-----------: | :-----------: | :--------------: | :--------------: |
> | Cora     |     81.40     |     75.03     |      80.80       |      80.88       |
> | Citeseer |     71.66     |     65.82     |      68.12       |      70.15       |
> | Arxiv    |     71.36     |     65.64     |      71.26       |      69.96       |
> | Flickr   |     47.12     |     47.02     |      47.11       |      47.08       |
> | Reddit   |     93.90     |     81.17     |      90.79       |      92.63       |
>
> > From the table, we observe the following:
> >
> > 1. removing the highest-degree nodes leads to the most significant drop in accuracy, yielding the lowest performance among the three methods;
> > 2. removing nodes with the simplest adjacency structure results in accuracy lower than Random only on the Citeseer dataset;
> > 3. and the results of the Random method generally fall between those of the other two strategies.
> >
> > The experimental results show that our conclusion, Nodes with high  capture richer structural information, is correct.
>
> ------
>
> **[Re-Weakness (3)] Lack of large-scale dataset validation**
>
> > We conduct baseline experiments and Time&Memory comparison experiments on the **Ogbn-Products** dataset to demonstrate the efficiency of our method.
> >
> > However, due to the large scale of the dataset, it will take some time to complete the experiments and present the results.

---

> ### Author Response · Authors · 2025-11-21
> **Response(2) to Reviewer j43U**
>
> **[Re-Weakness (4)] The source code or experimental settings to ensure reproducibility**
>
> > We will provide the source code and experimental environment configuration documentation in Github so that other researchers can successfully reproduce the experiment.
>
> ------
>
> **[Re-Weakness (5)] Dist is counter-intuitive to measure Similarity**
>
> > From our perspective, 'Dist'(distance) serves as a more objective measure for assessing similarity or relational closeness. We are also willing to revise the formulation accordingly.
>
> ------
>
> **[Re-Weakness (6) & Minor] The notation is unclear in places and other details**
>
> > Thank you very much for your valuable feedback.
> >
> > We confirm that all associated details in Minor and others have been fully addressed and the relevant papers were also discussed and cited in the revised paper.

---

### Official Review · Reviewer_c5bv · 2025-10-31

**Soundness:** 3
**Presentation:** 3
**Contribution:** 3
**Rating:** 6
**Confidence:** 3

**Summary:**

LIGHTGFC proposes a graph-free, three-stage condensation pipeline that embeds structure into features via multi-hop aggregation, trains an MLP expert, and then allocates condensed nodes per class using class-aware similarity with a lightweight projector. The objective combines label-adaptation (using the pretrained MLP) and prototype-alignment to preserve semantics and class discriminability without bi-level optimization over graphs and models. On transductive and inductive node-classification benchmarks, the method reports strong accuracy with very low training time (as little as seconds) and up to notable gains under tested ratios, highlighting efficiency without explicit condensed graph structures

**Strengths:**

1.  Efficient MLP-driven condensation that avoids nested optimization.
2.  Class-aware node allocation that better reflects class informativeness.
3.  Solid coverage of transductive and inductive datasets with promising results.

**Weaknesses:**

1. Potential loss of higher-order structure in graph-free condensation requires deeper validation.
2.  Need ablations on K-hop aggregation breadth, projector design, and class weighting robustness.
3.  Clarify fair-compute settings vs.\ recent baselines to substantiate SOTA claims.

**Questions:**

Scaling behavior and resource profiles across condensation ratios and graph sizes.
 Cross-backbone and inductive generalization with stability over seeds.
Analyze thresholds where structural loss limits downstream performance.

---

> ### Author Response · Authors · 2025-11-21
> **Response to Reviewer c5bv**
>
> We would like to express our gratitude for your constructive suggestions. We appreciate your recognition of the importance of this research topic.
>
> **[Re-Weakness(1)] Loss of higher-order structure in LightGFC**
>
> > Thank you so much for all your suggestions.
> >
> > The limitation of GNN layer count restricts its ability to learn higher-order structures. Therefore, the loss of higher-order structure in graph-free condensation has limited influence.
>
> ------
>
> **[Re-Weakness(2)] Ablation of relevant parameters**
>
> > We would like to clarify that the projector design and class weighting are not pre-set hyperparameters, and they are integral architectural component**s** of our model. So, they cannot be ablated in isolation.
> >
> > The 3-hop aggregation was empirically selected as the optimal configuration and we conducted an experiment to verify the impact of K-hop, which resulted as follows.
>
> | Dataset          | 1-hop | 2-hop | 3-hop | 4-hop | 5-hop |
> | ---------------- | ----- | ----- | ----- | ----- | ----- |
> | Cora (2.60%)     | 0.831 | 0.871 | 0.892 | 0.857 | 0.817 |
> | Citeseer (1.80%) | 0.766 | 0.820 | 0.836 | 0.796 | 0.791 |
> | Arxiv (0.05%)    | 0.653 | 0.665 | 0.674 | 0.667 | 0.663 |
> | Flickr (0.10%)   | 0.461 | 0.462 | 0.471 | 0.465 | 0.463 |
> | Reddit (0.05%)   | 0.916 | 0.921 | 0.923 | 0.920 | 0.917 |
>
> > K-hop parameters setting on different dataset with downstream GNN ACC.
> >
> > From the experimental results, we observe that:
> >
> > - 3-hop serves as the optimal setting across all datasets.
> > - For Cora, Citeseer, and Reddit—datasets with stronger feature representations—a smaller hop size (2-hop) is preferred.
> > - In contrast, Arxiv and Flickr, which have lower-dimensional feature vectors, benefit from aggregating information over more hops.
>
> ------
>
> **[Re-Weakness(3)] Fair-compute baseline setting**
>
> > In our baseline comparison experiment of graph condensation, Fair computation contains:
> >
> > (1) **dataset split**: we used the training/test ratio or the mask code provided with the OBG to ensure consistency;
> >
> > (2) **condensation ratio**: we have chosen the compression ratio (Nodes Per Class) first proposed by GCond, and many methods like SFGC, CGC, and GCPA all use these compression ratios;
> >
> > (3) **downstream GNN model**: we used classic PyG GCN model architecture, without any modifications, and set 10 random seeds for multiple downstream task tests.
>
> ------
>
> ------
>
> **[Re-Question(1)] Scaling behavior and resource profiles across condensation ratios and graph sizes.**
>
> > For the mainstream settings of compression ratios, there are two common approaches: (i) defining the compression ratio based on fixed numbers of nodes per class after compression (e.g., 10, 20, etc.), and (ii) directly specifying a ratio without constraining the exact number of nodes.
> >
> > Each approach has its own advantages and disadvantages, but since the relative relationships across classes remain unchanged, their impact on downstream GNN performance is minimal.
>
> ------
>
> **[Re-Question(2)] Cross-backbone and inductive generalization with stability over seeds.**
>
> > **Cross-backbone Validation & Generalization**: In Table 2, we evaluate the effectiveness of the condensed graph across multiple backbones, including SGC, SAGE, and JKNet. The results show that the performance is both stable and strong.
> >
> > **Stability**: Each experiment was repeated with 10 different random seeds to ensure the stability of the algorithm.
>
> ------
>
> **[Re-Question(3)] Analyze thresholds where structural loss limits downstream performance.**
>
> > To evaluate the influence of structural loss on downstream tasks, we designed a dedicated experiment.
> >
> > To retain part of the topological information, we generate edges within the compressed graph based on feature similarity and then measure the downstream task accuracy under this setting.
>
> | Method                          | Cora (2.60%) | Citeseer (1.80%) | Arxiv (0.05%) | Flickr (0.10%) | Reddit (0.05%) |
> | ------------------------------- | ------------ | ---------------- | ------------- | -------------- | -------------- |
> | LightGFC                        | 0.892        | 0.836            | 0.674         | 0.471          | 0.923          |
> | LightGFC (with simple topology) | 0.803        | 0.811            | 0.641         | 0.471          | 0.917          |
>
> > From the experimental results, we find that for most datasets, structure-free data already achieves strong performance. Even when adding intuitive topological information that is expected to be helpful, the improvement is limited and can sometimes even degrade performance.

---

### Official Review · Reviewer_HU9W · 2025-11-01

**Soundness:** 2
**Presentation:** 2
**Contribution:** 3
**Rating:** 2
**Confidence:** 4

**Summary:**

The authors propose a novel graph condensation algorithm using a three staged pipeline which involves an aggregation step over the graph to generate a structure free proto graph and then using that to train an MLP, learn an informative distribution of labels and then do label to node condensation to generate the final graph using a learnable projector matrix.
This technique is very light weight relying on MLPs, provokes some interesting thoughts on using different label distributions and results in an extremely fast and lightweight state-of-the-art condensation algorithm with amazing results on basic evaluation benchmark.

**Strengths:**

The algorithm uses very few hyper-parameters, is stable with those hyperparamters and also consumes significantly less time and memory than other techniques.

I believe the concept of experimenting with different label distributions and finding “better” has a lot of value for condensation. But this process needs metrics, justification and robustness analysis.

Components S2 and S3 stimulate interesting thoughts for the broader graph community and how simple experiments can lead to strong results as shown by table 1 and table 2.

The numbers in table 1 and 2 are impressive and improvements are significant

**Weaknesses:**

Lack of code raises serious reproducibility concerns. Number of things need to be specified for benchmarking time and accuracy. The hyperparameters of the base GNN class used, number of cores on which the process is running/ parallelism exploited, libraries and environment used are needed to check the stats.

Not all the techniques mentioned in related works are benchmarked. In fact, few techniques mentioned in table 1 are missing in table 2 and timing and memory plots.

Table 2 does not have standard deviations. This is not a proper presentation method. Numbers have to be reliable and statistically significant.

Reasoning needs to be a “bigger component” than experimentation. The statistics in table 1 and 2 are very interesting. For eg:
LightGFC outperforms the full dataset on cora and citeseer and very close to full in reddit but not in others, what can be the reason?
Multiple techniques like Bonsai, protostack, lightgfc etc are outperforming the full dataset, is it correct to then rank them solely on basis of performance on these datasets? Are these small datasets then even worth benchmarking?
In GIN for cora, the performance suddenly drops drastically, the only difference between gcn and gin is aggregation, this performance I believe is even worse than randomly selecting nodes. How is that possible?

The challenge C2 is not “intrinsic” to these algorithms. In fact, any label distribution can be defined at start and the final distribution can be made to mimic it. The most faithful choice however,the occams razor, is the original distribution as it would most closely resemble real world data. I believe this is one reason why a random sample also usually gives good condensation results (although this benchmark is absent in your tables)

I think the ProtoAggregation phase is a very common precompute strategy. It does not feel like a novel component you have introduced. The major strength is in finding “better label distributions” and label to node condensation. But again how is your label distribution better is nowhere written. A comparison is needed with original distribution where better results are received.

I believe in conclusion a lot of experimentation is not strongly evaluated and lot of experimentation is missing. There is also not a lot of theory developed as to why this method should be able to outperform full training and achieve the numbers presented in table 1 and 2

Minor typo, is M of line 242 same as P of line 244? Otherwise where is M used? How is P learnt?

**Questions:**

Can there be an algorithm section which formally lists the steps of the algorithm, the paper organisation makes it hard to keep track of where various things are happening and what is used where. For Eg: What is the use of L_total and where is used?

Why are there no numbers for the full dataset when running different models like GIN,SAGE etc?

Why are different compression ratios taken for different datasets? This reference “Bonsai: Gradient-free Graph Condensation for Node Classification” mentions use of size compression metric as opposed to node compression used by GCond. Different sized compressed outputs would trivially store different information and hence show different performance. Can we have a size based comparison

What does - - - in Flickr for Protostack mean?

Can we also include analysis on larger sized datasets like MAG240M and ogbn-papers?

Some important and recent techniques like GDEM and EXGC are excluded, can they be included?
1) Graph Distillation with Eigenbasis Matching
2)EXGC: Bridging Efficiency and Explainability in Graph Condensation

For the different datasets, can we look at how different the output label distributions of your model are as compared to training data

Can some more clarity be given on ablation techniques? How are the metric computed when various components are disabled? For eg, how is compressed data formed with only PAlign. Also you mention Idx1-3 disable each of the components once but your ticks and crosses show something else?

Can we have a runtime and memory analysis of the different stages as well? This can show if there is scope of improvement via parallelism

---

> ### Author Response · Authors · 2025-11-21
> **Response(1) to Reviewer HU9W**
>
> Thank you for your thoughtful review and constructive suggestions. We greatly appreciate your recognition of our method’s motivation and potential.
>
> **[Re-Weakness(1)] Code for reproducibility**
>
> > Due to the need to prepare supplementary experiments during the Rebuttal phase, we will send the anonymous link to AC for your reference.
>
> ------
>
> **[Re-Weakness(2)] Baseline in Discussion and Experiment**
>
> > Due to the page limit, we conduct experiments by selecting the algorithms most highly related and representative to our proposed method, LightGFC. We conduct all the comparison experiments, covering EXGC and GDEM, and more detailed results are listed in the appendix.
>
> ------
>
> **[Re-Weakness(3)] Std. in Table2**
>
> > In the revised paper, we will add the standard deviation to Table 2.
>
> ------
>
> **[Re-Weakness(4)] Performance across whole dataset; the worth benchmark of small dataset; the generalization ability of GIN**
>
> > Regarding why the method can even surpass the performance on the original datasets:
> >
> > (1) The essence of distribution matching is to approximate the feature-based classification behavior of GCN. LightGFC further improves node feature representations through **the MLP and Lightweight Projector**, leading to better optimization.
> >
> > Table 4 in the appendix describes the specific attributes of each dataset. As a graph-free method, LightGFC is particularly effective on datasets with complex feature vectors, which explains its strongest performance on Cora and Citeseer. It also performs well on Arxiv and Reddit, though less so on Flickr, which shows that LightGFC is not only advantageous on small datasets.
> >
> > (2) Within a limited dataset, **more effective condensation method should extract as much important information as possible to enhance downstream performance.** Therefore, on these datasets, the performance across the whole dataset serves as an indicator of how well a method is able to capture and utilize the latent information in the original even small graph.
> >
> > (3) GIN uses sum aggregation, whereas GCN uses mean aggregation. Since the Label-aware loss in the Lightweight Class-to-Node Condensation stage essentially performs **class center averaging**, this mismatch leads to weaker performance on GIN, particularly on feature dominant datasets like Cora.
>
> ------
>
> **[Re-Weakness(5)] Random method & The reason to adjust distribution**
>
> > Random sampling is the simplest approach and has been consistently shown to be the weakest baseline in many comparative studies. Therefore, we did not include it in the originally submitted paper.
> >
> > Thinks to your suggestion, and we are willing to provide these results as follow.
>
> | Method   | Cora (2.6%) | Citeseer (1.8%) | Arxiv (0.25%) | Flickr (0.50%) | Reddit (0.10%) |
> | -------- | ----------- | --------------- | ------------- | -------------- | -------------- |
> | Random   | 72.8 ± 1.1  | 64.2 ± 1.7      | 57.3 ± 1.1    | 44.0 ± 0.4     | 58.0 ± 2.2     |
> | LightGFC | 90.6 ± 0.9  | 82.8 ± 0.3      | 68.1 ± 0.3    | 47.3 ± 0.5     | 91.8 ± 0.4     |
>
> > **The feature distribution obtained directly from the original graph partition is insufficient to meet the training requirements of downstream GNNs.** Therefore, distribution adjustment is necessary to improve performance.
>
> ------
>
> **[Re-Weakness(6)] The novelty of ProtoAggregation  & Better Label Distribution**
>
> > (1) The novelty of ProtoAggregation as follow:
> >
> > **(i)** After ProtoAggregation, the data aggregate as a structure-free data, making it suitable for MLP pretraining.
> >
> > **(ii)** The node weights are computed by comparing the ProtoAggregation features with the original features, enabling the Class-to-Node Condensation to perform the compression.
>
> > (2) Similarity-based weight can clearly identify influence nodes, which modify a lot in stage Multi-hop Topology Aggregation, thereby adjusting the overall proportion. **Increasing the proportion of influential labels is how to make the label distribution better.**
>
> ------
>
> **[Re-Weakness(7)] Theory evaluate result & L_total**
>
> > (1) **The goal of graph condensation is to facilitate downstream GNN training**, rather than to simply replicate the distribution and feature of the original graph. This is why baseline methods, including CGC, GCPA, DisCo, ProStack, and Bonsai, have outperformed the whole-dataset baseline.
>
> > (2) During the feature optimization with the pretrained MLP, the enhanced features also lead to the removal of some redundant edges.
> >
> > This mechanism effectively **performs feature enhancement by  focusing on essential nodes**, which allows our condensed graph to achieve performance comparable to, or even better than, the full dataset.
>
> ------
>
> **[Re-Weakness(8)] Minor typo of M**
>
> > The symbol in equations have been corrected.

---

> ### Author Response · Authors · 2025-11-21
> **Response(2) to Reviewer HU9W**
>
> **[Re-Question(1)] The steps of the algorithm & the use of L_total**
>
> ```reStructuredText
> Alogrithm 1: The framework of LightGFC
> Input: Original graph G={A, X, Y}, N denotes the numbers of graph G, C denotes the label of Graph G, Lightweight Projector M
> Output: Condensation graph G'={X', Y'}
> /* Proto-structural Aggregation */
> Aggregate feature information from neighboring nodes
> for i = 1,...,N
> 	h_i = Iulti-hop Topology Aggregation(x_i, a_i, X)
> Proto-graph-free Data H = [h_1,...,h_N]
> /* MLP-driven Structure-free Pretraining */
> for i = 1,...,N
> 	optimaztion Mlp with Loss(MLP(h_i),y_i)
> Obtain MLP*
> /* Lightweight Class-to-node Condensation */
> for j = 1,...,C
> 	for i = 1,...,(n in j) // the number of nodes in label j
> 		w_i = Simility computation Dist(h^j_i, x^j_i)
> H',Y' = Class-to-Node Condensation(H, Y, W)
> X' = Optimzation (MLP*, M)
> Retrun G'={X', H'}
> ```
>
> > (The pseudocode implemented using Markdown text is shown above. We will include more elegant pseudocode written in LaTeX in the appendix.)
>
> > (1) The detailed algorithm of LightGFC is shown in Algorithm  1.
> >
> > In the **Proto-structural Aggregation** stage, nodes aggregate features from their neighbors through the adjacency matrix to obtain $\text H$. The resulting $\text H$ is then used to train an MLP classifier. For each class, we compute the spatial distance between the current feature vector $\text h$ and the original feature vector $\text x$ as a similarity measure.
> >
> > After normalization with weighted aggregation, we obtain the weight $\text w$ for each node. Finally, we compress the graph using $\text w$, producing the reduced feature matrix $\text H'$ and label vector $\text Y'$. Afterward, the Lightweight Projector is applied to optimize $\text H'$, yielding the final condensed graph $(\text X', \text Y')$.
> >
> > (2) L_total is used in **Stage S3: Lightweight Class-to-node Condensation**, when optimizing the **Lightweight Projector M**.
>
> ------
>
> **[Re-Question(2)] Without full dataset & different compression ratios**
>
> > (1) Compared to the baseline experiments, the generalization performance experiments focused more on testing multiple downstream GNN models. Therefore, when the number of pages was limited, a single dataset was chosen with a compression ratio that preserved a more GNN models.
> > We will add generalization performance test experiments for full dataset.
>
> > (2) For the mainstream settings of compression ratios, there are two common approaches:
> >
> > (i) defining the compression ratio based on fixed numbers of nodes per class after compression (e.g., 10, 20, etc.), and the ratio is obtained by dividing the fixed numbers by the original numbers.
> >
> > For example, in the Cora dataset with 7 classes, if we retain 5 nodes per class, we obtain a total of 35 nodes. The compression ratio is therefore 35/2708, which is approximately 1.3%.
> >
> > (ii) directly specifying a ratio without constraining the exact number of nodes. Previous graph-free and distribution-matching approaches have typically adopted the first type of condensation ratio setting, and for consistency, we followed the same choice.
> >
> > We chose the ratio(i) instead of Bonsai’s because **the number of nodes per class is multiples of 5(5, 10, 20) that makes it more convenient for the Class-to-Node Condensation stage to synthesize nodes through weight.**
>
> ------
>
> **[Re-Question(3)] Without data in Prostack Flickr & Large sized dataset**
>
> > Since ProStack has not released its code and cannot be reliably reproduced, we did not include results on the Flickr dataset.
> >
> > We will conduct additional experiments on Obgn-products, which is also a large-scale graph dataset. Running Obgn-products consumes more time, so we will analyze the results and include them in the appendix within the next two to three days.

---

> > ### Comment · Reviewer_HU9W · 2025-11-23
> > **Clarifications**
> >
> > I dont understand why you cant incorporate one specific dataset (named Flickr) when you have implemented ProStack. Either all work on ProStack should have been unreliable or Flickr should work.
> > I believe size based compression ratios are "fairer" in informational theoretical sense as highlighted by Bonsai.
> > I dont find your justification "We chose the ratio(i) instead of Bonsai’s because the number of nodes per class is multiples of 5(5, 10, 20) that makes it more convenient for the Class-to-Node Condensation stage to synthesize nodes through weight." to  uphold the integrity of experimentation. It has become a popular practice, but I believe it is not the best and most fair practice.

---

> ### Author Response · Authors · 2025-11-21
> **Response(3) to Reviewer HU9W**
>
> **[Re-Question(4)] Comparison GDEM and EXGC**
>
> > EXGC first compresses the original graph into an EM framework and then iteratively updates the synthetic nodes, thereby improving training efficiency.
> >
> > GDEM avoids reliance on GNN-specific information by matching the synthetic and original graphs, and further incorporates class-level constraints to achieve efficient and highly generalizable graph condensation.
>
> > As graph condensation methods, they can be included in discussion and comparison. However, as we can see from the above introduction, their relevance to LightGFC, which focuses on **distribution matching** and **graph-free data**, is relatively weak. Therefore, they were not initially chosen, and relevant content will be added in the appendix later.
>
> **[Re-Question(5)] Difference between origin and condensation Distribution**
>
> >In the appendix, we provide a bar chart. It uses labels as the X-axis and node proportions as the Y-axis to illustrate the changes in distribution before and after compression for LightGFC and baselines.
> >From the figure, we can observe that, unlike baseline methods that allocate an equal number of compressed nodes to each class, LightGFC slightly adjusts the node proportions across labels.
>
> **[Re-Question(6)] Explanation of Ablation & PAlign**
>
> > (1) The overall framework of our method is relatively simple, and the ablation study contains two parts: compression and optimization. For the optimization ablation, we disable the two components separately by setting their corresponding hyperparameters to zero. **For the compression ablation, we replace our module with the standard K-Center method.**
> >
> > (2) For the variants Id1–Id4 in Table 3:
> >
> > ​	(i) Id1 and Id2 disable the prototype-aware feature alignment and label-aware model adaptation modules individually by setting their corresponding hyperparameters to zero, preventing these components from participating in the Projector’s optimization.
> >
> > ​	(ii) Id3 disables both components simultaneously, meaning that the compressed graph is directly used for GNN evaluation without any adjustment.
> >
> > ​	(iii) To further validate our proposed compression process, Id4 replaces our compression module with the K-Center method while retaining the full Projector learning procedure.(In the original version of the paper, there was a typo error in the configuration of Id1. The correct setting should be ✓ × ✓.)
>
> ------
>
> **[Re-Question(7)] Runtime and Memory of different stages**
>
> > The following table shows the **runtime and memory** usage of each LightGFC component:
>
> | Dataset        | Pre-training MLP | Condensation Graph | Lightweight Projector optimization |
> | -------------- | ---------------- | ------------------ | ---------------------------------- |
> | Cora(2.6%)     | 0.5s / 72MB      | 1.2s / 372MB       | 2.0s / 637MB                       |
> | Citeseer(1.8%) | 0.7s / 150MB     | 1.6s / 475MB       | 2.7s / 804MB                       |
> | Arxiv(0.25%)   | 11.4s / 302MB    | 17.8s / 665MB      | 30.6s / 1212MB                     |
> | Flickr(0.5%)   | 9.6s / 368MB     | 20.3s / 702MB      | 25.8s / 1283MB                     |
> | Reddit(0.1%)   | 10.2s / 973MB    | 18.5s / 1863MB     | 26.6s / 3444MB                     |
>
> > Runtime identifies the 'Lightweight Projector Optimization' stage as the dominant computational bottleneck, consuming nearly 50% of the total execution time on large-scale datasets (Reddit and Arxiv). Additionally, memory consumption correlates positively with dataset complexity, reaching a peak of 3.4GB for Reddit in this final stage.
> >
> > The program's sequential dependency on memory makes it difficult to run in parallel.

---

> ### Comment · Reviewer_HU9W · 2025-11-23
> **More clarifications**
>
> Does your reply to point 4 inherently mean that there is poor (theoretically backed) generalisability of your work? Other GIN results apart from one highlighted dont look _that bad_.
> I am still not convinced of what are the novel contributions you have made to ProtoAggregation phase. Many other papers perform similar techniques and put them as part of preprocessing.
> I would also humbly urge the authors to share the codebase with all reviewers. This would allow:
> a)everyone to test the codebase
> b) everyone can discuss the codebase openly if there are any issues.

---

> ### Author Response · Authors · 2025-12-01
> **Response(4) to Reviewer HU9W**
>
> We sincerely thank you for your response again, which has provided valuable insights and new perspectives for improving our work.
>
> **[Re-More clarifications] Generalizability of work ; The novel contribution of ProtoAggregation; Share the codebase**
>
> > (1) As shown in Table 2, the results demonstrate that our method exhibits strong generalization capability across various datasets and architectures. The only exception appears on GIN. **This is primarily because the label-aware loss used in the Lightweight Class-to-Node Condensation stage essentially performs class-center averaging.** Such averaging mismatches the characteristics of GIN, whose expressive power relies heavily on discriminative feature patterns. This issue becomes more evident on feature-dominant datasets like Cora, leading to the relatively weaker performance observed on GIN.
> >
> > (2) **The ProtoAggregation module in LightGFC serves two novel purposes** that have not been explored in prior work. (i)It enables the aggregation of graph-free data, which facilitates both MLP pre-training and condensation. (ii)The aggregated feature matrix provides the foundation for computing similarities and weights. Taken together, these two roles demonstrate that our use of the ProtoAggregation component is indeed innovative.
> >
> > (3) We have already provided the link to the anonymous repository to the AC(Area Chairs), who will share it with the reviewers for further inspection.
>
> ------
>
> **[Re-Clarifications] Lack dataset Flickr of ProStack; size based compression ratios are "fairer"in Bonsai**
>
> > (1) Since ProStack has not released its source code and experimental environment, we were unable to successfully reproduce its results. In the ProStack paper, only four datasets are reported, excluding Flickr; therefore, the Flickr result is unavailable.
>
> > (2) The two types of compression ratios are constructed based on different design philosophies. Therefore, the difference between them does not imply that one ratio is inherently more correct or superior than the other.
> >
> > Below are the results obtained under the same condensation ratios as Bonsai, ensuring a fair and consistent comparison across all methods.
>
> | Dataset  | Ratio(%) |    Bonsai    |   LightGFC   |
> | :------: | :------: | :----------: | :----------: |
> |   Cora   |   0.5    |   83.9±0.4   | **88.4±0.2** |
> |   Cora   |   1.0    |   85.8±0.2   | **88.9±0.5** |
> |   Cora   |   3.0    |   86.4±0.2   | **90.2±0.7** |
> | Citeseer |   0.5    |   77.0±0.2   | **80.7±0.9** |
> | Citeseer |   1.0    |   77.0±0.3   | **82.0±0.1** |
> | Citeseer |   3.0    |   75.9±0.3   | **83.1±0.3** |
> |  Arxiv   |   0.5    |   58.5±0.2   | **67.7±0.1** |
> |  Arxiv   |   1.0    |   58.4±0.1   | **67.2±0.4** |
> |  Arxiv   |   3.0    |   64.3±0.1   | **66.9±0.3** |
> |  Flickr  |   0.5    | **48.7±0.3** |   47.3±0.5   |
> |  Flickr  |   1.0    | **49.1±0.2** |   47.4±0.2   |
> |  Flickr  |   3.0    | **49.7±0.3** |   47.4±0.5   |
> |  Reddit  |   0.5    |   80.3±0.5   | **91.3±0.5** |
> |  Reddit  |   1.0    |   85.7±0.1   | **90.9±0.0** |
> |  Reddit  |   3.0    |   88.9±0.1   | **90.2±0.7** |
>
> > As highlighted in **bold**, the results show that LightGFC achieves consistently better condensation performance on all datasets except Flickr.
> >
> > This demonstrates that, under Bonsai’s condensation ratios, LightGFC remains highly effective and competitive across diverse graph benchmarks.

---

### Official Review · Reviewer_tx4B · 2025-11-04

**Soundness:** 2
**Presentation:** 3
**Contribution:** 2
**Rating:** 4
**Confidence:** 3

**Summary:**

This paper proposes LIGHTGFC (LIGHTweight Graph-Free Condensation with MLP-driven optimization), a novel method that condenses large-scale graph data into a structure-free node set in a simple, accurate, yet highly efficient manner. The approach consists of three stages—proto-structural aggregation, MLP-driven structural-free pretraining, and lightweight class-to-node condensation—to embed structural information, align representations, and generate representative nodes. Extensive experiments demonstrate that LIGHTGFC achieves state-of-the-art accuracy across multiple benchmarks while requiring minimal training time (as little as 2.0s), highlighting both its effectiveness and efficiency.

**Strengths:**

1. The proposed method effectively eliminates bi-level optimization inefficiency through an MLP-driven structure-free condensation process.
2. It introduces a clear three-stage framework that preserves structural and semantic information while maintaining lightweight computation.
3. Extensive experiments confirm strong performance gains and remarkable efficiency, achieving state-of-the-art accuracy with minimal training time.

**Weaknesses:**

1. Although this paper outperforms the baselines, many of the main contributions mentioned (especially in the abstract) resemble ideas already proposed in existing methods such as SimGC [1], GCPA [2], and CGC [3]. Given the strong performance, the authors should further clarify the novelty of their approach compared to these works.
2. Some highly related baselines, such as [1], are not discussed.
3. It is unclear why the baseline CGC-X is not included in Figure 2 for comparison.

[1] Zhenbang Xiao, Yu Wang, Shunyu Liu, Huiqiong Wang, Mingli Song, and Tongya Zheng. "Simple graph condensation." In Joint European Conference on Machine Learning and Knowledge Discovery in Databases, pp. 53-71. Cham: Springer Nature Switzerland, 2024.
[2] Yuan Li, Jun Hu, Zemin Liu, Bryan Hooi, Jia Chen, and Bingsheng He. "Adapting Precomputed Features for Efficient Graph Condensation." In Forty-second International Conference on Machine Learning.
[3] Xinyi Gao, Guanhua Ye, Tong Chen, Wentao Zhang, Junliang Yu, and Hongzhi Yin. "Rethinking and accelerating graph condensation: A training-free approach with class partition." In Proceedings of the ACM on Web Conference 2025, pp. 4359-4373. 2025

**Questions:**

See Weaknesses.

---

> ### Author Response · Authors · 2025-11-21
> **Response to Reviewer tx4B**
>
> We sincerely thank you for your comprehensive review and insightful comments. We have made every effort to address your concerns and detailed our revisions below.
>
> **[Re-Weakness(1)] Comparison with baselines in Novelty and Contribution**
>
> > The difference among the distribution matching methods as follow:
>
> | Components                       |              SimGC              |               CGC               |                             GCPA                             |             LightGFC              |
> | -------------------------------- | :-----------------------------: | :-----------------------------: | :----------------------------------------------------------: | :-------------------------------: |
> | **Feature propagation**          | $H^{(k)}=\hat{A}^{(k)} \cdot X$ | $H^{(k)}=\hat{A}^{(k)} \cdot X$ | $H^{(k)}=(1- \alpha) \hat{A}^{(k)} \cdot H^{(k-1)} + \alpha \cdot H^{(0)}$ | $H^{(k-1)}=\hat{A}^{(k)} \cdot X$ |
> | **Model training**               |           train-free            |           train-free            |                          Adaptation                          |       Lightweight Projector       |
> | **Distribution optimization**    |                /                |                /                |                              /                               |     Class weight optimization     |
> | **Condensation graph synthesis** |           Clustering            |   Class partition with conf.    |                     Adaptation synthesis                     |    Similarity weight synthesis    |
>
> > We compare following aspects: Feature propagation, Model training, Distribution optimization and Condensation graph synthesis.
> >
> > (1) **Feature propagation**: This method aggregates features from neighboring nodes using $H^{(k-1)}=\hat{A}^{(k)} \cdot X$ to modify the feature vector of the node itself.
> >
> > Unlike the other three methods primarily focus on data augmentation, our feature aggregation serves as the foundation for all subsequent stages. Different aggregated features generate similarity relationships, which are then used to assign weights and adjust the distribution.
> >
> > (2) **Model training**: different from Adaptation of GCPA which used to synthesis graph,  **Lightweight projector** directly optimizes the feature matrix of condensation graph. So the learning effect of lightweight projector is more direct and concrete.
> >
> > (3) **Distribution optimization & Condensation graph synthesis**: SimGC, CGC, and GCPA directly obtain the feature distribution after calculation, but they do not have a learning and adjustment process. LightGFC uses the **Similarity weight synthesis** component in **Class-to-Node Condensation** to compress node features based on weight synthesis, adjusting the distribution and feature vectors of label nodes.
> >
> > Therefore, we obtain a compressed subgraph that is more consistent with GNN operations.
>
> ------
>
> **[Re-Weakness(2)] Discussion SimGC**
>
> > By pretraining an SGC to provide hierarchical semantic representations, SimGC optimizes only the condensed graph so that it matches the original graph's distribution at every layer. And it aligns the output logits and applies smooth regularization to enhance structural consistency. As a result, SimGC obtains a high-quality condensed graph at extremely low cost.
> >
> > The key differences between SimGC and LightGFC are as follows:
> >
> > (1) SimGC directly computes the feature matrix of the condensed graph, whereas LightGFC relies on a Lightweight Projector for this purpose.
> >
> > (2) SimGC constructs the condensed graph through clustering, while LightGFC generates node feature vectors using similarity-weight synthesis to obtain the final subgraph.
> >
> > The baseline experiment result of SimGC will be added to the appendix.
>
> ------
>
> ** [Re-Weakness(3)] CGC-X not in Figure2(Time&Memory comparison experiment)  **
>
> > The baseline CGC-X is a train-free model of graph condensation. It does not require time for training GNN and only involves mathematical operations. So it is more comparable in Time&Memory experiment without CGC-X for other methods.

---

### Author Response · Authors · 2025-12-02
**Common Response to All Reviewers and AC**

We sincerely thank you (the AC) for taking the time to reassess our submission. We would like to thank all reviewers positive comments to our work and we hope all our response could address their questions.



We thank all reviewers for their thorough review and valuable suggestions. We are delighted that our contributions have been positively acknowledged, including:

1. The efficient MLP-driven graph-free condensation framework that eliminates bi-level optimization inefficiency (**Reviewer tx4B**, **Reviewer c5bv**), along with its stability and low memory consumption (**Reviewer HU9W**);

2. The novel strategy to break label distribution constraints and class-aware node allocation (**Reviewer j43U**, **Reviewer c5bv**, **Reviewer HU9W**);

3. A clear three-stage framework that preserves semantic information (**Reviewer tx4B**); and

4. Comprehensive empirical evaluation confirming strong performance gains and remarkable efficiency over a wide range of baselines (**Reviewer tx4B**, **Reviewer HU9W**, **Reviewer j43U**, **Reviewer c5bv**).

In particular, we are greatly encouraged by the remarks of **Reviewer HU9W**, who found the experimental results ‘impressive’ and noted that our components ‘stimulate interesting thoughts for the broader graph community’.

Additionally, we greatly appreciate the positive feedback from **Reviewer tx4B** on our framework achieving ‘state-of-the-art accuracy with minimal training time’, as well as **Reviewer j43U**’s assessment of our strategy as a ‘novel contribution to the field of graph condensation’.

These comments encourage us to continue our efforts in advancing this field of research.



### **Summary**

We present a lightweight framework that achieves highly efficient and high-performance graph condensation in a graph-free manner.

Our approach decomposes the bi-level optimization bottleneck and redesigns the condensed label distribution through a similarity-based adjustment strategy, which leads to consistent performance improvements.

All additional materials and supplementary results mentioned in the rebuttal will be included in the updated appendix of the paper.

---

### Meta-Review · Area_Chair_rPCN · 2026-01-14

**Summary:**

The reviewers’ concerns are mostly related to (1) Novelty; (2) Hyperparameter and Component Validity; (3) Performance Anomalies.

**Reviewer Concerns:**

(1) Novelty\
The reviewers believe that the main contribution of the paper is highly similar to existing methods like SimGC, GCPA. And the main component ProtoAggregation cannot be seen as a novel component as it is common used.
The authors renamed the aggregation of SGC as Proto-structural aggregation, the essence of which is still just low-pass filter. This cannot be considered as an innovation.

(2)Hyperparameter and Component Validity\
The reviewers questioned that when setting the loss weight to zero (effectively disabling the optimization) improves the performance, which is counter-intuitive.
The authors could not convincingly explain it, implying the proposed module might be redundant or detrimental.

(3) Performance Anomalies\
The reviewers questioned why the performance trained on Cora and Citeseer is better than trained on the whole dataset, and why Gin has a low performance on Cora, even worse than random selection.
The authors claimed that MLP and Projector optimize the feature and Gin uses sum aggregation which is not compatible with mean aggregation used by LightGFC. I believe overfitting happens in the training process on Cora and Citeseer. And if the condensation method cannot be applied to GIN, it's a lack of persuasiveness.

**Reviewer Scores:**

I think no reviewer will change their scores as the main concerns are not addressed during the rebuttal process.

---

### Decision · Program_Chairs · 2026-01-26

Reject